# *h*-detach: Modifying the LSTM Gradient Towards Better Optimization

**Bhargav Kanuparthi**[*,1]**, Devansh Arpit**[*,1]**, Giancarlo Kerg**[1]**, Nan Rosemary Ke**[1]**,
Ioannis Mitliagkas**[1] **& Yoshua Bengio**[1,2]
[1]Montreal Institute for Learning Algorithms (MILA), Canada
[2]CIFAR Senior Fellow
*Authors contributed equally
{bhargavkanuparthi25,devansharpit}@gmail.com

## Abstract

Recurrent neural networks are known for their notorious exploding and vanishing gradient problem (EVGP). This problem becomes more evident in tasks where the information needed to correctly solve them exist over long time scales, because EVGP prevents important gradient components from being back-propagated adequately over a large number of steps. We introduce a simple stochastic algorithm (*h*-detach) that is specific to LSTM optimization and targeted towards addressing this problem. Specifically, we show that when the LSTM weights are large, the gradient components through the linear path (cell state) in the LSTM computational graph get suppressed. Based on the hypothesis that these components carry information about long term dependencies (which we show empirically), their suppression can prevent LSTMs from capturing them. Our algorithm[1] prevents gradients flowing through this path from getting suppressed, thus allowing the LSTM to capture such dependencies better. We show significant improvements over vanilla LSTM gradient based training in terms of convergence speed, robustness to seed and learning rate, and generalization using our modification of LSTM gradient on various benchmark datasets.

## 1 Introduction

Recurrent Neural Networks (RNNs) (Rumelhart et al. (1986); Elman (1990)) are a class of neural network architectures used for modeling sequential data. Compared to feed-forward networks, the loss landscape of recurrent neural networks are much harder to optimize. Among others, this difficulty may be attributed to the exploding and vanishing gradient problem (Hochreiter, 1991; Bengio et al., 1994; Pascanu et al., 2013) which is more severe for recurrent networks and arises due to the highly ill-conditioned nature of their loss surface. This problem becomes more evident in tasks where training data has dependencies that exist over long time scales.

Due to the aforementioned optimization difficulty, variants of RNN architectures have been proposed that aim at addressing these problems. The most popular among such architectures that are used in a wide number of applications include long short term memory (LSTM, Hochreiter & Schmidhuber (1997)) and gated recurrent unit (GRU, Chung et al. (2014)) networks, which is a variant of LSTM with forget gates (Gers et al., 1999). These architectures mitigate such difficulties by introducing a *linear temporal path* that allows gradients to flow more freely across time steps. Arjovsky et al. (2016) on the other hand try to address this problem by parameterizing a recurrent neural network to have unitary transition matrices based on the idea that unitary matrices have unit singular values which prevents gradients from exploding/vanishing.

Among the aforementioned RNN architectures, LSTMs are arguably most widely used (for instance they have more representational power compared with GRUs (Weiss et al., 2018)) and it remains a hard problem to optimize them on tasks that involve long term dependencies. Examples of such tasks are copying problem (Bengio et al., 1994; Pascanu et al., 2013), and sequential MNIST (Le

---

[1]Our code is available at https://github.com/bhargav104/h-detach.

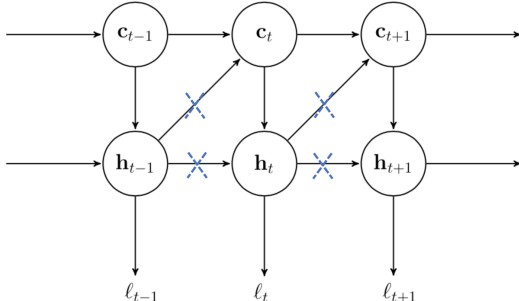

Figure 1: The computational graph of a typical LSTM. Here we have omitted the inputs $\mathbf{x}_i$ for convenience. The top horizontal path through the cell state units $\mathbf{c}_t$s is the linear temporal path which allows gradients to flow more freely over long durations. The dotted blue crosses along the computational paths denote the stochastic process of blocking the flow of gradients though the $\mathbf{h}_t$ states (see Eq 2) during the back-propagation phase of LSTM. We call this approach $h$-detach.

et al., 2015), which are designed in such a way that the only way to produce the correct output is for the model to retain information over long time scales.

The goal of this paper is to introduce a simple trick that is specific to LSTM optimization and improves its training on tasks that involve long term dependencies. To achieve this goal, we write out the full back-propagation gradient equation for LSTM parameters and split the composition of this gradient into its components resulting from different paths in the unrolled network. We then show that when LSTM weights are large in magnitude, the gradients through the linear temporal path (cell state) get suppressed (recall that this path was designed to allow smooth gradient flow over many time steps). We show empirical evidence that this path carries information about long term dependencies (see section 3.5) and hence gradients from this path getting suppressed is problematic for such tasks. To fix this problem, we introduce a simple stochastic algorithm that in expectation scales the individual gradient components, which prevents the gradients through the linear temporal path from being suppressed. In essence, the algorithm stochastically prevents gradient from flowing through the $h$-state of the LSTM (see figure 1), hence we call it $h$-detach. Using this method, we show improvements in convergence/generalization over vanilla LSTM optimization on the copying task, transfer copying task, sequential and permuted MNIST, and image captioning.

## 2 Proposed Method: $h$-detach

We begin by reviewing the LSTM roll-out equations. We then derive the LSTM back-propagation equations and by studying its decomposition, identify the aforementioned problem. Based on this analysis we propose a simple stochastic algorithm to fix this problem.

### 2.1 Long Short Term Memory Networks

LSTM is a variant of traditional RNNs that was designed with the goal of improving the flow of gradients over many time steps. The roll-out equations of an LSTM are as follows,

$$\mathbf{c}_t = \mathbf{f}_t \odot \mathbf{c}_{t-1} + \mathbf{i}_t \odot \mathbf{g}_t \tag{1}$$
$$\mathbf{h}_t = \mathbf{o}_t \odot tanh(\mathbf{c}_t) \tag{2}$$

where $\odot$ denotes point-wise product and the gates $\mathbf{f}_t$, $\mathbf{i}_t$, $\mathbf{o}_t$ and $\mathbf{g}_t$ are defined as,

$$\mathbf{g}_t = tanh(\mathbf{W}_{gh}\mathbf{h}_{t-1} + \mathbf{W}_{gx}\mathbf{x}_t + \mathbf{b}_g) \tag{3}$$
$$\mathbf{f}_t = \sigma(\mathbf{W}_{fh}\mathbf{h}_{t-1} + \mathbf{W}_{fx}\mathbf{x}_t + \mathbf{b}_f) \tag{4}$$
$$\mathbf{i}_t = \sigma(\mathbf{W}_{ih}\mathbf{h}_{t-1} + \mathbf{W}_{ix}\mathbf{x}_t + \mathbf{b}_i) \tag{5}$$
$$\mathbf{o}_t = \sigma(\mathbf{W}_{oh}\mathbf{h}_{t-1} + \mathbf{W}_{ox}\mathbf{x}_t + \mathbf{b}_o) \tag{6}$$

Here $\mathbf{c}_t$ and $\mathbf{h}_t$ are the cell state and hidden state respectively. Usually a transformation $\phi(\mathbf{h}_T)$ is used as the output at time step $t$ (Eg. next word prediction in language model) based on which we can compute the loss $\ell_t := \ell(\phi(\mathbf{h}_t))$ for that time step.

An important feature of the LSTM architecture is the linear recursive relation between the cell states $\mathbf{c}_t$ as shown in Eq. 1. This linear path allows gradients to flow easily over long time scales. This however is one of the components in the full composition of the LSTM gradient. As we will show next, the remaining components that are a result of the other paths in the LSTM computational graph are polynomial in the weight matrices $\mathbf{W}_{gh}, \mathbf{W}_{fh}, \mathbf{W}_{ih}, \mathbf{W}_{oh}$ whose order grows with the number of time steps. These terms cause an imbalance in the order of magnitude of gradients from different paths, thereby suppressing gradients from linear paths of LSTM computational graph in cases where the weight matrices are large.

## 2.2 BACK-PROPAGATION EQUATIONS FOR LSTM

In this section we derive the back-propagation equations for LSTM network and by studying its composition, we identify a problem in this composition. The back-propagation equation of an LSTM can be written in the following form.

**Theorem 1** *Fix $w$ to be an element of the matrix $\mathbf{W}_{gh}, \mathbf{W}_{fh}, \mathbf{W}_{ih}, \mathbf{W}_{oh}, \mathbf{W}_{gx}, \mathbf{W}_{fx}, \mathbf{W}_{ix}$ or $\mathbf{W}_{ox}$. Define,*

$$\mathbf{A}_t = \begin{bmatrix} \mathbf{F}_t & \mathbf{0}_n & diag(\mathbf{k}_t) \\ \tilde{\mathbf{F}}_t & \mathbf{0}_n & diag(\tilde{\mathbf{k}}_t) \\ \mathbf{0}_n & \mathbf{0}_n & \mathbf{Id}_n \end{bmatrix} \quad \mathbf{B}_t = \begin{bmatrix} \mathbf{0}_n & \psi_t & \mathbf{0}_n \\ \mathbf{0}_n & \tilde{\psi}_t & \mathbf{0}_n \\ \mathbf{0}_n & \mathbf{0}_n & \mathbf{0}_n \end{bmatrix} \quad \mathbf{z}_t = \begin{bmatrix} \frac{d\mathbf{c}_t}{dw} \\ \frac{d\mathbf{h}_t}{dw} \\ \mathbf{1}_n \end{bmatrix} \tag{7}$$

*Then $\mathbf{z}_t = (\mathbf{A}_t + \mathbf{B}_t)\mathbf{z}_{t-1}$. In other words,*

$$\mathbf{z}_t = (\mathbf{A}_t + \mathbf{B}_t)(\mathbf{A}_{t-1} + \mathbf{B}_{t-1})\ldots(\mathbf{A}_2 + \mathbf{B}_2)\mathbf{z}_1 \tag{8}$$

*where all the symbols used to define $\mathbf{A}_t$ and $\mathbf{B}_t$ are defined in notation 1 in appendix.*

To avoid unnecessary details, we use a compressed definitions of $\mathbf{A}_t$ and $\mathbf{B}_t$ in the above statement and write the detailed definitions of the symbols that constitute them in notation 1 in appendix. Nonetheless, we now provide some intuitive properties of the matrices $\mathbf{A}_t$ and $\mathbf{B}_t$.

The matrix $\mathbf{A}_t$ contains components of parameter's full gradient that arise due to the cell state (linear temporal path) described in Eq. (1) (top most horizontal path in figure 1). Thus the terms in $\mathbf{A}_t$ are a function of the LSTM gates and hidden and cell states. Note that all the gates and hidden states $\mathbf{h}_t$ are bounded by definition because they are a result of sigmoid or tanh activation functions. The cell state $\mathbf{c}_t$ on the other hand evolves through a linear recursive equation shown in Eq. (1). Thus it can grow at each time step by at most $\pm 1$ (element-wise) and its value is bounded by the number of time steps $t$. Thus given a finite number of time steps and finite initialization of $\mathbf{c}_0$, the values in matrix $\mathbf{A}_t$ are bounded.

The matrix $\mathbf{B}_t$ on the other hand contains components of parameter's full gradient that arise due to the remaining paths. The elements of $\mathbf{B}_t$ are a linear function of the weights $\mathbf{W}_{gh}, \mathbf{W}_{fh}, \mathbf{W}_{ih}, \mathbf{W}_{oh}$. Thus the magnitude of elements in $\mathbf{B}_t$ can become very large irrespective of the number of time steps if the weights are very large. This problem becomes worse when we multiply $\mathbf{B}_t$s in Eq. (8) because the product becomes polynomial in the weights which can become unbounded for large weights very quickly as the number of time steps grow.

Thus based on the above analysis, we identify the following problem with the LSTM gradient: when the LSTM weights are large, the gradient component through the cell state paths ($\mathbf{A}_t$) get suppressed compared to the gradient components through the other paths ($\mathbf{B}_t$) due to an imbalance in gradient component magnitudes. We recall that the linear recursion in the cell state path was introduced in the LSTM architecture (Hochreiter & Schmidhuber, 1997) as an important feature to allow gradients to flow smoothly through time. As we show in our ablation studies (section 3.5), this path carries information about long term dependencies in the data. Hence it is problematic if the gradient components from this path get suppressed.

## 2.3 $h$-DETACH

We now propose a simple fix to the above problem. Our goal is to manipulate the gradient components such that the components through the cell state path ($\mathbf{A}_t$) do not get suppressed when the components through the remaining paths ($\mathbf{B}_t$) are very large (described in the section 2.2). Thus

it would be helpful to multiply $\mathbf{B}_t$ by a positive number less than 1 to dampen its magnitude. In Algorithm 1 we propose a simple trick that achieves this goal. A diagrammatic form of algorithm 1 is shown in Figure 1. In simple words, our algorithm essentially blocks gradients from flowing through each of the $\mathbf{h}_t$ states independently with a probability $1 - p$, where $p \in [0, 1]$ is a tunable hyper-parameter. Note the subtle detail in Algorithm 1 (line 9) that the loss $\ell_t$ at any time step $t$ is a function of $\mathbf{h}_t$ which is not detached.

---

**Algorithm 1** Forward Pass of $h$-detach Algorithm

---

 1: **INPUT:** $\{\mathbf{x}_t\}_{t=1}^{T}$, $\mathbf{h}_0$, $\mathbf{c}_0$, $p$
 2: $\ell = 0$
 3: **for** $1 \leq t \leq T$ **do**
 4:     **if** bernoulli($p$)==1 **then**
 5:         $\tilde{\mathbf{h}}_{t-1} \leftarrow$ stop-gradient($\mathbf{h}_{t-1}$)
 6:     **else**
 7:         $\tilde{\mathbf{h}}_{t-1} \leftarrow \mathbf{h}_{t-1}$
 8:     $\mathbf{h}_t, \mathbf{c}_t \leftarrow \text{LSTM}(\mathbf{x}_t, \tilde{\mathbf{h}}_{t-1}, \mathbf{c}_{t-1})$                                          (Eq. 1- 6)
 9:     $\ell_t \leftarrow \text{loss}(\phi(\mathbf{h}_t))$
10:     $\ell \leftarrow \ell + \ell_t$
11: **return** $\ell$

---

We now show that the gradient of the loss function resulting from the LSTM forward pass shown in algorithm 1 has the property that the gradient components arising from $\mathbf{B}_t$ get dampened.

**Theorem 2** *Let* $\mathbf{z}_t = [\frac{d\mathbf{c}_t}{dw}^T; \frac{d\mathbf{h}_t}{dw}^T; \mathbf{1}_n^T]^T$ *and* $\tilde{\mathbf{z}}_t$ *be the analogue of* $\mathbf{z}_t$ *when applying* $h$-detach *with probability* $1 - p$ *during back-propagation. Then,*

$$\tilde{\mathbf{z}}_t = (\mathbf{A}_t + \xi_t \mathbf{B}_t)(\mathbf{A}_{t-1} + \xi_{t-1}\mathbf{B}_{t-1})\ldots(\mathbf{A}_2 + \xi_2\mathbf{B}_2)\tilde{\mathbf{z}}_1$$

*where* $\xi_t, \xi_{t-1}, \ldots, \xi_2$ *are i.i.d. Bernoulli random variables with probability* $p$ *of being 1, and* $w$, $\mathbf{A}_t$ *and* $\mathbf{B}_t$ *and are same as defined in theorem 1.*

The above theorem shows that by stochastically blocking gradients from flowing through the $\mathbf{h}_t$ states of an LSTM with probability $1 - p$, we stochastically drop the $\mathbf{B}_t$ term in the gradient components. The corollary below shows that in expectation, this results in dampening the $\mathbf{B}_t$ term compared to the original LSTM gradient.

**Corollary 1** $\mathbb{E}_{\xi_2,\ldots,\xi_t}[\tilde{\mathbf{z}}_t] = (\mathbf{A}_t + p\mathbf{B}_t)(\mathbf{A}_{t-1} + p\mathbf{B}_{t-1})\ldots(\mathbf{A}_2 + p\mathbf{B}_2)\tilde{\mathbf{z}}_1$

Finally, we note that when training LSTMs with $h$-detach, we reduce the amount of computation needed. This is simply because by stochastically blocking the gradient from flowing through the $h_t$ hidden states of LSTM, less computation needs to be done during back-propagation through time (BPTT).

## 3 EXPERIMENTS

### 3.1 COPYING TASK

This task requires the recurrent network to memorize the network inputs provided at the first few time steps and output them in the same order after a large time delay. Thus the only way to solve this task is for the network to capture the long term dependency between inputs and targets which requires gradient components carrying this information to flow through many time steps.

We follow the copying task setup identical to Arjovsky et al. (2016) (described in appendix). Using their data generation process, we sample 100,000 training input-target sequence pairs and 5,000 validation pairs. We use cross-entropy as our loss to train an LSTM with hidden state size 128 for a maximum of 500-600 epochs. We use the ADAM optimizer with batch-size 100, learning rate 0.001 and clip the gradient norms to 1.

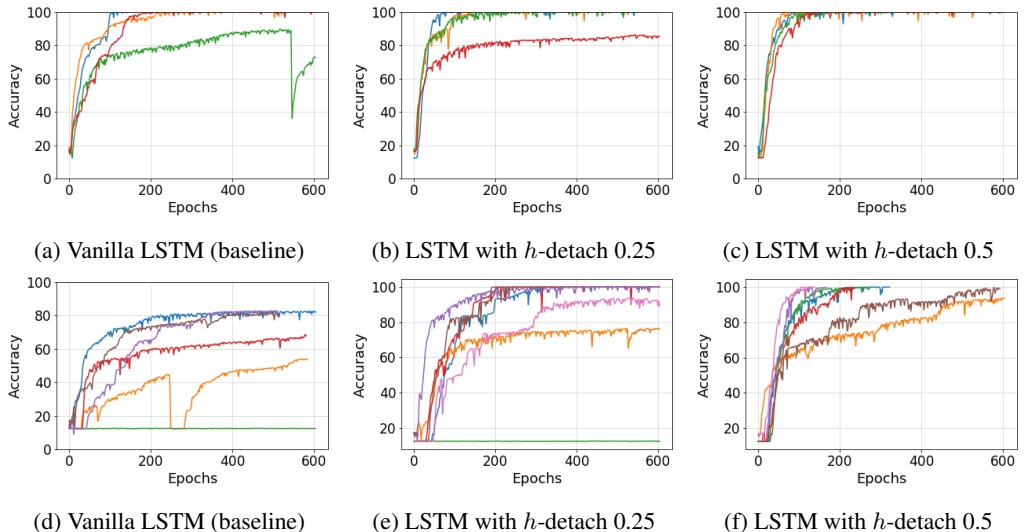

(a) Vanilla LSTM (baseline)      (b) LSTM with $h$-detach 0.25      (c) LSTM with $h$-detach 0.5

(d) Vanilla LSTM (baseline)      (e) LSTM with $h$-detach 0.25      (f) LSTM with $h$-detach 0.5

Figure 2: Validation accuracy curves during training on copying task using vanilla LSTM (left) and LSTM with $h$-detach with probability 0.25 (middle) and 0.5 (right). **Top row** is delay $T = 100$ and **bottom row** is delay $T = 300$. Each plot contains multiple runs with different seeds. We see that for $T = 100$, even the baseline LSTM is able to reach $\sim 100\%$ accuracy for most seeds and the only difference we see between vanilla LSTM and LSTM with $h$-detach is in terms of convergence. $T = 300$ is a more interesting case because it involves longer term dependencies. In this case we find that $h$-detach leads to faster convergence and achieves $\sim 100\%$ validation accuracy while being more robust to the choice of seed.

Figure 2 shows the validation accuracy plots for copying task training for $T = 100$ (top row) and $T = 300$ (bottom row) without $h$-detach (left), and with $h$-detach (middle and right). Each plot contains runs from the same algorithm with multiple seeds to show a healthy sample of variations using these algorithms. For $T = 100$ time delay, we see both vanilla LSTM and LSTM with $h$-detach converge to $100\%$ accuracy. For time delay 100 and the training setting used, vanilla LSTM is known to converge to optimal validation performance (for instance, see Arjovsky et al. (2016)). Nonetheless, we note that $h$-detach converges faster in this setting. A more interesting case is when time decay is set to 300 because it requires capturing longer term dependencies. In this case, we find that LSTM training without $h$-detach achieves a validation accuracy of $\sim 82\%$ at best while a number of other seeds converge to much worse performance. On the other hand, we find that using $h$-detach with detach probabilities 0.25 and 0.5 achieves the best performance of $100\%$ and converging quickly while being reasonably robust to the choice of seed.

## 3.2 TRANSFER COPYING TASK

Having shown the benefit of $h$-detach in terms of training dynamics, we now extend the challenge of the copying task by evaluating how well an LSTM trained on data with a certain time delay generalizes when a larger time delay is used during inference. This task is referred as the transfer copying task (Hochreiter & Schmidhuber, 1997). Specifically, we train the LSTM architecture on copying task with delay $T = 100$ without $h$-detach and with $h$-detach with probability 0.25 and 0.5. We then evaluate the accuracy of the trained model for each setting for various values of $T > 100$. The results are shown in table 1. We find that the function learned by LSTM when trained with $h$-detach generalize significantly better on longer time delays during inference compared with the LSTM trained without $h$-detach.

## 3.3 SEQUENTIAL MNIST

This task is a sequential version of the MNIST classification task (LeCun & Cortes, 2010). In this task, an image is fed into the LSTM one pixel per time step and the goal is to predict the label after the last pixel is fed. We consider two versions of the task: one is which the pixels are read in order

| T | VanillaLSTM | $h$-detach 0.5 | $h$-detach 0.25 |
|---|---|---|---|
| 200 | 64.85 | 74.79 | **90.72** |
| 400 | 48.17 | 54.91 | **77.76** |
| 500 | 43.03 | 52.43 | **74.68** |
| 1000 | 28.82 | 43.54 | **63.19** |
| 2000 | 19.48 | 34.34 | **51.83** |
| 5000 | 14.58 | 24.55 | **42.35** |

Table 1: Accuracy on transfer copying task. We find that the generalization of LSTMs trained with $h$-detach is significantly better compared with vanilla LSTM training when tested on time delays longer that what the model is trained on ($T = 100$).

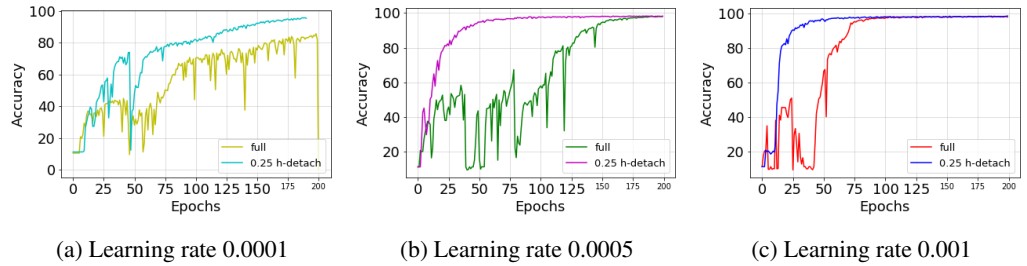

(a) Learning rate 0.0001    (b) Learning rate 0.0005    (c) Learning rate 0.001

Figure 3: Validation accuracy curves of LSTM training on pixel by pixel MNIST. Each plot shows LSTM training with and without $h$-detach for different values of learning rate. We find that $h$-detach is both more robust to different learing rates and converges faster compared to vanilla LSTM training. Refer to the Fig. 6 in appendix for validation curves on multiple seeds.

(from left to right and top to bottom), and one where all the pixels are permuted in a random but fixed order. We call the second version the permuted MNIST task or pMNIST in short. The setup used for this experiment is as follows. We use 50000 images for training, 10000 for validation and 10000 for testing. We use the ADAM optimizer with different learning rates– 0.001,0.0005 and 0.0001, and a fixed batch size of 100. We train for 200 epochs and pick our final model based on the best validation score. We use an LSTM with 100 hidden units. For $h$-detach, we do a hyper-parameter search on the detach probability in $\{0.1, 0.25, 0.4, 0.5\}$. For both pixel by pixel MNIST and pMNIST, we found the detach hyper-parameter of $0.25$ to perform best on the validation set for both MNIST and pMNIST.

On the sequential MNIST task, both vanilla LSTM and training with $h$-detach give an accuracy of $98.5\%$. Here, we note that the convergence of our method is much faster and is more robust to the different learning rates of the ADAM optimizer as seen in Figure 3. Refer to appendix (figure 6) for experiments with multiple seeds that shows the robustness of our method to initialization.

In the pMNIST task, we find that training LSTM with $h$-detach gives a test accuracy of $92.3\%$ which is an improvement over the regular LSTM training which reaches an accuracy of $91.1\%$. A detailed comparison of test performance with existing algorithms is shown in table 2.

| Method | MNIST | pMNIST |
|---|---|---|
| Vanilla LSTM | 98.5 | 91.1 |
| SAB (Ke et al., 2018) | - | 94.2 |
| iRNN Le et al. (2015) | 97.0 | 82.0 |
| uRNN (Arjovsky et al., 2016) | 95.1 | 91.4 |
| Zoneout (Krueger et al., 2016) | - | **93.1** |
| IndRNN (Li et al., 2018) | **99** | **96** |
| $h$-detach (ours) | 98.5 | 92.3 |

Table 2: A comparison of test accuracy on pixel by pixel MNIST and permuted MNIST (pMNIST) with existing methods.

Table 3: Test performance on image captioning task on MS COCO dataset using metrics BLEU 1 to 4, METEOR, and CIDEr (higher values are better for all metrics). We re-implement both Show&Tell (Vinyals et al., 2015) and Soft Attention (Xu et al., 2015) and train the LSTM in these models with and without $h$-detach.

| Models | B-1 | B-2 | B-3 | B-4 | METEOR | CIDEr |
|---|---|---|---|---|---|---|
| DeepVS (Karpathy & Fei-Fei, 2015) | 62.5 | 45.0 | 32.1 | 23.0 | 19.5 | 66.0 |
| ATT-FCN (You et al., 2016) | 70.9 | 53.7 | 40.2 | 30.4 | 24.3 | — |
| Show & Tell (Vinyals et al., 2015) | — | — | — | 27.7 | 23.7 | 85.5 |
| Soft Attention (Xu et al., 2015) | 70.7 | 49.2 | 34.4 | 24.3 | 23.9 | — |
| Hard Attention (Xu et al., 2015) | 71.8 | 50.4 | 35.7 | 25.0 | 23.0 | — |
| MSM (Yao et al., 2017) | 73.0 | 56.5 | 42.9 | 32.5 | 25.1 | 98.6 |
| Adaptive Attention (Lu et al., 2017) | **74.2** | **58.0** | **43.9** | **33.2** | **26.6** | **108.5** |
| TwinNet (Serdyuk et al., 2018) | | | | | | |
| *No attention, Resnet152* | 72.3 | 55.2 | 40.4 | 29.3 | 25.1 | 94.7 |
| *Soft Attention, Resnet152* | 73.8 | 56.9 | 42.0 | 30.6 | 25.2 | 97.3 |
| *No attention, Resnet152* | | | | | | |
| Show&Tell (Our impl.) | 71.7 | 54.4 | 39.7 | 28.8 | 24.8 | 93.0 |
| + $h$-detach (0.25) | **72.9** | **55.8** | **41.7** | **31.0** | **25.1** | **98.0** |
| *Attention, Resnet152* | | | | | | |
| Soft Attention (Our impl.) | 73.2 | 56.3 | 41.4 | 30.1 | 25.3 | 96.6 |
| + $h$-detach (0.4) | **74.7** | **58.1** | **44.0** | **33.1** | **26.0** | **103.3** |

## 3.4 IMAGE CAPTIONING

We now evaluate $h$-detach on an image captioning task which involves using an RNN for generating captions for images. We use the Microsoft COCO dataset (Lin et al., 2014) which contains 82,783 training images and 40,504 validation images. Since this dataset does not have a standard split for training, validation and test, we follow the setting in Karpathy & Fei-Fei (2015) which suggests a split of 80,000 training images and 5,000 images each for validation and test set.

We use two models to test our approach– the Show&Tell encoder-decoder model (Vinyals et al., 2015) which does not employ any attention mechanism, and the 'Show, Attend and Tell' model (Xu et al., 2015), which uses soft attention. For feature extraction, we use the 2048-dimensional last layer feature vector of a residual network (Resnet He et al. (2015)) with 152 layers which was pre-trained on ImageNet for image classification. We use an LSTM with 512 hidden units for caption generation. We train both the Resnet and LSTM models using the ADAM optimizer (Kingma & Ba, 2014) with a learning rate of $10^{-4}$ and leave the rest of the hyper-parameters as suggested in their paper. We also perform a small hyperparameter search where we find the optimial value of the $h$-detach parameter. We considered values in the set $\{0.1, 0.25, 0.4, 0.5\}$ and pick the optimal value based on the best validation score. Similar to Serdyuk et al. (2018), we early stop based on the validation CIDEr scores and report BLEU-1 to BLEU-4, CIDEr, and Meteor scores.

The results are presented in table 3. Training the LSTM with $h$-detach outperforms the baseline LSTM by a good margin for all the metrics and produces the best BLEU-1 to BLEU-3 scores among all the compared methods. Even for the other metrics, except for the results reported by Lu et al. (2017), we beat all the other methods reported. We emphasize that compared to all the other reported methods, $h$-detach is extremely simple to implement and does not add any computational overhead (in fact reduces computation).

## 3.5 ABLATION STUDIES

In this section, we first study the effect of removing gradient clipping in the LSTM training and compare how the training of vanilla LSTM and our method get affected. Getting rid of gradient clipping would be insightful because it would confirm our claim that stochastically blocking gradients through the hidden states $h_t$ of the LSTM prevent the growth of gradient magnitude. We train both models on pixel by pixel MNIST using ADAM without any gradient clipping. The validation accuracy curves are reported in figure 4 for two different learning rates. We notice that removing gradient clipping causes the Vanilla LSTM training to become extremely unstable. $h$-detach on the

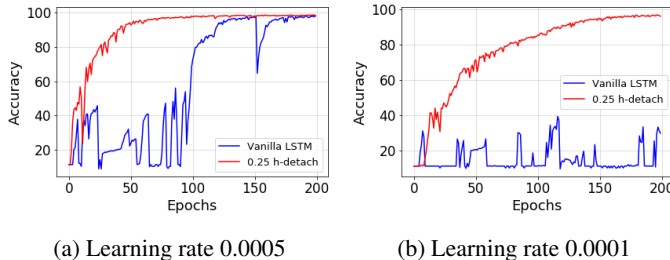

(a) Learning rate 0.0005       (b) Learning rate 0.0001

Figure 4: The effect of removing gradient clipping from vanilla LSTM training vs. LSTM trained with $h$-detach on pixel by pixel MNIST dataset. Refer to Fig. 8 in appendix for experiments with multiple seeds.

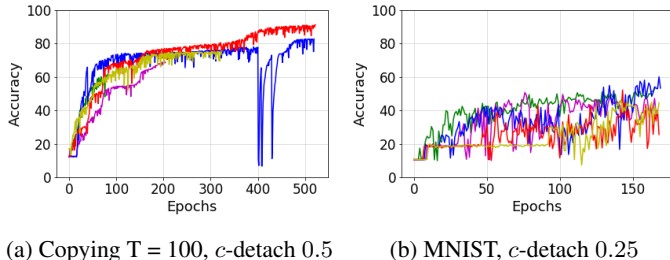

(a) Copying T = 100, $c$-detach 0.5       (b) MNIST, $c$-detach 0.25

Figure 5: Validation accuracy curves for copying task T=100 (left) and pixel by pixel MNIST (right) using LSTM such that gradient is stochastically blocked through the cell state (the probability of detaching the cell state in this experiment is mentioned in sub-titles.). Blocking gradients from flowing through the cell state path of LSTM ($c$-detach) leads to significantly worse performance compared even to vanilla LSTM on tasks that requires long term dependencies. This suggests that the cell state path carry information about long term dependencies.

other hand seems robust to removing gradient clipping for both the learning rates used. Additional experiments with multiple seeds and learning rates can be found in figure 8 in appendix.

Second, we conduct experiments where we stochastically block gradients from flowing through the cell state $\mathbf{c}_t$ instead of the hidden state $\mathbf{h}_t$ and observe how the LSTM behaves in such a scenario. We refer detaching the cell state as $c$-detach. The goal of this experiment is to corroborate our hypothesis that the gradients through the cell state path carry information about long term dependencies. Figure 5 shows the effect of $c$-detach (with probabilities shown) on copying task and pixel by pixel MNIST task. We notice in the copying task for $T = 100$, learning becomes very slow (figure 5 (a)) and does not converge even after 500 epochs, whereas when not detaching the cell state, even the Vanilla LSTM converges in around 150 epochs for most cases for T=100 as shown in the experiments in section 3.1. For pixel by pixel MNIST (which involves 784 time steps), there is a much larger detrimental effect on learning as we find that none of the seeds cross $60\%$ accuracy at the end of training (Figure 5 (b)). This experiment corroborates our hypothesis that gradients through the cell state contain important components of the gradient signal as blocking them worsens the performance of these models when compared to Vanilla LSTM.

## 4 RELATED WORK

Capturing long term dependencies in data using recurrent neural networks has been long known to be a hard problem (Hochreiter, 1991; Bengio et al., 1993). Therefore, there has been a considerable amount of work on addressing this issue. Prior to the invention of the LSTM architecture (Hochreiter & Schmidhuber, 1997), another class of architectures called NARX (nonlinear autoregressive models with exogenous) recurrent networks (Lin et al., 1996) was popular for tasks involving long term dependencies. More recently gated recurrent unit (GRU) networks (Chung et al., 2014) was proposed that adapts some favorable properties of LSTM while requiring fewer parameters. Other recent recurrent architecture designs that are aimed at preventing EVGP can be found in Zhang et al.

(2018), Jose et al. (2017) and Li et al. (2018). Work has also been done towards better optimization for such tasks (Martens & Sutskever, 2011; Kingma & Ba, 2014). Since vanishing and exploding gradient problems (Hochreiter, 1991; Bengio et al., 1994) also hinder this goal, gradient clipping methods have been proposed to alleviate this problem (Tomas, 2012; Pascanu et al., 2013). Yet another line of work focuses on making use of unitary transition matrices in order to avoid loss of information as hidden states evolve over time. Le et al. (2015) propose to initialize recurrent networks with unitary weights while Arjovsky et al. (2016) propose a new network parameterization that ensures that the state transition matrix remains unitary. Extensions of the unitary RNNs have been proposed in Wisdom et al. (2016), Mhammedi et al. (2016) and Jing et al. (2016). Very recently, Ke et al. (2018) propose to learn an attention mechanism over past hidden states and sparsely back-propagate through paths with high attention weights in order to capture long term dependencies. Trinh et al. (2018) propose to add an unsupervised auxiliary loss to the original objective that is designed to encourage the network to capture such dependencies. We point out that our proposal in this paper is orthogonal to a number of the aforementioned papers and may even be applied in conjunction to some of them. Further, our method is specific to LSTM optimization and reduces computation relative to the vanilla LSTM optimization which is in stark contrast to most of the aforementioned approaches which increase the amount of computation needed for training.

## 5 DISCUSSION AND FUTURE WORK

In section 3.5 we showed that LSTMs trained with $h$-detach are stable even without gradient clipping. We caution that while this is true, in general the gradient magnitude depends on the value of detaching probability used in $h$-detach. Hence for the general case, we do not recommend removing gradient clipping.

When training stacked LSTMs, there are two ways in which $h$-detach can be used: 1) detaching the hidden state of all LSTMs simultaneously for a given time step $t$ depending on the stochastic variable $\xi_t$) stochastically detaching the hidden state of each LSTM separately. We leave this for future work.

$h$-detach stochastically blocks the gradient from flowing through the hidden states of LSTM. In corollary 1, we showed that in expectation, this is equivalent to dampening the gradient components from paths other than the cell state path. We especially chose this strategy because of its ease of implementation in current auto-differentiation libraries. Another approach to dampen these gradient components would be to directly multiply these components with a dampening factor. This feature is currently unavailable in these libraries but may be an interesting direction to look into. A downside of using this strategy though is that it will not reduce the amount of computation similar to $h$-detach (although it will not increase the amount of computation compared with vanilla LSTM either). Regularizing the recurrent weight matrices to have small norm can also potentially prevent the gradient components from the cell state path from being suppressed but it may also restrict the representational power of the model.

Given the superficial similarity of $h$-detach with dropout, we outline the difference between the two methods. Dropout randomly masks the hidden units of a network during the forward pass (and can be seen as a variant of the stochastic delta rule (Hanson, 1990)). Therefore, a common view of dropout is training an ensemble of networks (Warde-Farley et al., 2013). On the other hand, our method does not mask the hidden units during the forward pass. It instead randomly blocks the gradient component through the h-states of the LSTM only during the backward pass and does not change the output of the network during forward pass. More specifically, our theoretical analysis shows the precise behavior of our method: the effect of $h$-detach is that it changes the update direction used for descent which prevents the gradients through the cell state path from being suppressed.

We would also like to point out that even though we show improvements on the image captioning task, it does not fit the profile of a task involving long term dependencies that we focus on. We believe the reason why our method leads to improvements on this task is that the gradient components from the cell state path are important for this task and our theoretical analysis shows that $h$-detach prevents these components from getting suppressed compared with the gradient components from the other paths. On the same note, we also tried our method on language modeling tasks but did not notice any benefit.

## 6 CONCLUSION

We proposed a simple stochastic algorithm called $h$-detach aimed at improving LSTM performance on tasks that involve long term dependencies. We provided a theoretical understanding of the method using a novel analysis of the back-propagation equations of the LSTM architecture. We note that our method reduces the amount of computation needed during training compared to vanilla LSTM training. Finally, we empirically showed that $h$-detach is robust to initialization, makes the convergence of LSTM faster, and/or improves generalization compared to vanilla LSTM (and other existing methods) on various benchmark datasets.

ACKNOWLEDGMENTS

We thank Stanisław Jastrzębski, David Kruger and Isabela Albuquerque for helpful discussions. DA was supported by IVADO.

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

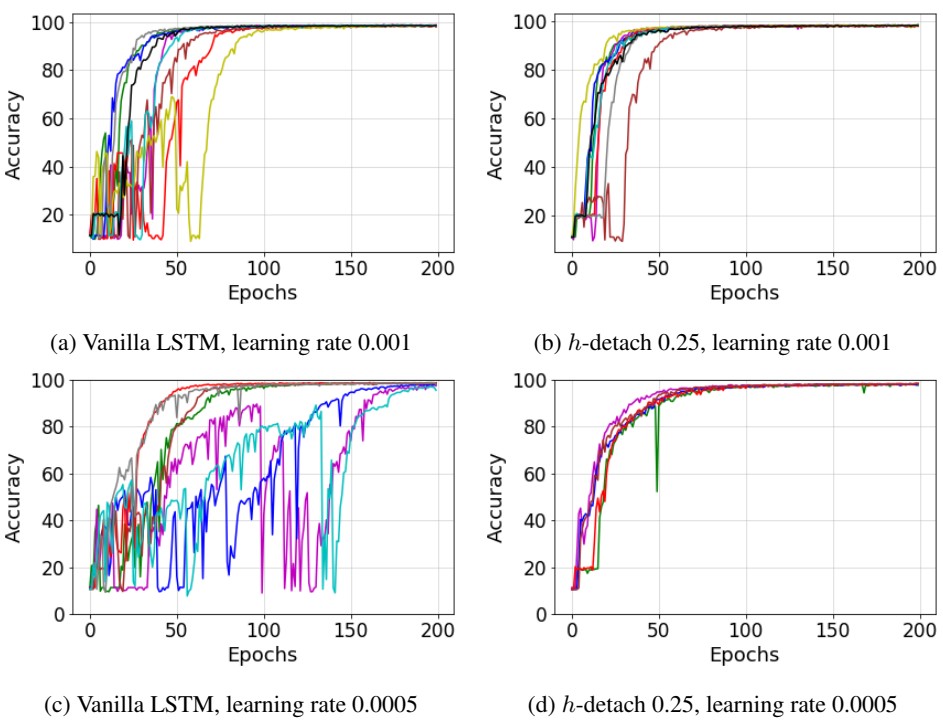

(a) Vanilla LSTM, learning rate 0.001

(b) $h$-detach 0.25, learning rate 0.001

(c) Vanilla LSTM, learning rate 0.0005

(d) $h$-detach 0.25, learning rate 0.0005

Figure 6: Validation accuracy curves on pixel by pixel MNIST dataset with vanilla LSTM training and LSTM training with $h$-detach with various values of learning rate and initialization seeds.

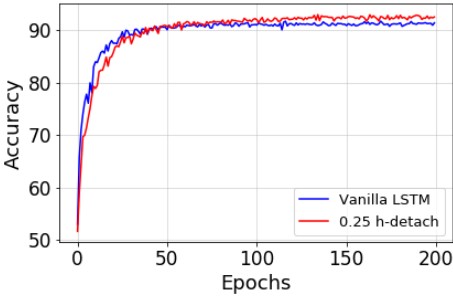

Figure 7: Validation accuracy curves on pMNIST dataset with vanilla LSTM training and LSTM training with $h$-detach.

## APPENDIX

## A ADDITIONAL INFORMATION

Copying Experiment setup - We define 10 tokens, $\{a_i\}_{i=0}^9$. The input to the LSTM is a sequence of length $T + 20$ formed using one of the ten tokens at each time step. Input for the first 10 time steps are sampled i.i.d. (uniformly) from $\{a_i\}_{i=0}^7$. The next $T - 1$ entries are set to $a_8$, which constitutes a delay. The next single entry is $a_9$, which represents a delimiter, which should indicate to the algorithm that it is now required to reproduce the initial 10 input tokens as output. The remaining 10 input entries are set to $a_8$. The target sequence consists of $T + 10$ repeated entries of $a_8$, followed by the first 10 entries of the input sequence in exactly the same order.

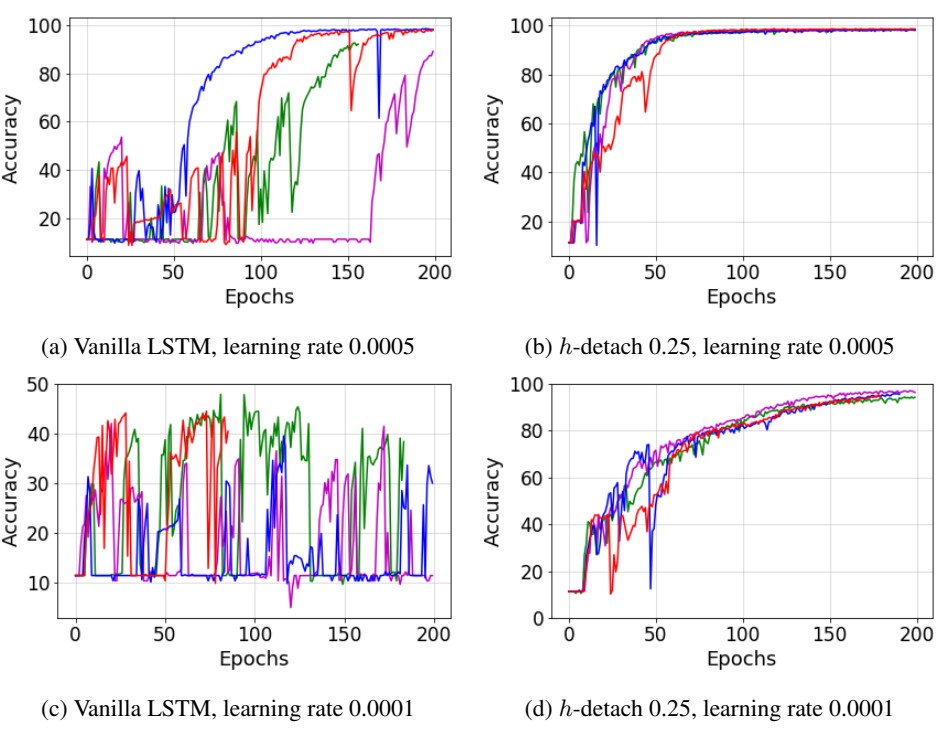

(a) Vanilla LSTM, learning rate 0.0005

(b) $h$-detach 0.25, learning rate 0.0005

(c) Vanilla LSTM, learning rate 0.0001

(d) $h$-detach 0.25, learning rate 0.0001

Figure 8: The effect of removing gradient clipping during optimization. Validation accuracy curves on pixel by pixel MNIST dataset with vanilla LSTM training and LSTM training with $h$-detach with various values of learning rate and initialization seeds. LSTM training using $h$-detach is both significantly more stable and robust to initialization when removing gradient clipping compared with vanilla LSTM training.

# B DERIVATION OF BACK-PROPAGATION EQUATION FOR LSTM

Let us recall the equations from an LSTM

$$\mathbf{o}_t = \sigma \left( \mathbf{W}_o [\mathbf{h}_{t-1}, \mathbf{x}_t]^T + \mathbf{b}_o \right)$$

$$\mathbf{i}_t = \sigma \left( \mathbf{W}_i [\mathbf{h}_{t-1}, \mathbf{x}_t]^T + \mathbf{b}_i \right)$$

$$\mathbf{g}_t = \tanh \left( \mathbf{W}_g [\mathbf{h}_{t-1}, \mathbf{x}_t]^T + \mathbf{b}_g \right)$$

$$\mathbf{f}_t = \sigma \left( \mathbf{W}_f [\mathbf{h}_{t-1}, \mathbf{x}_t]^T + \mathbf{b}_f \right)$$

$$\mathbf{h}_t = \mathbf{o}_t \odot \tanh(\mathbf{c}_t)$$

$$\mathbf{c}_t = \mathbf{f}_t \odot \mathbf{c}_{t-1} + \mathbf{i}_t \odot \mathbf{g}_t$$

Here $\odot$ denotes the element-wise product, also called the Hadamard product. $\sigma$ denotes the sigmoid activation function. $\mathbf{W}_o = [\mathbf{W}_{oh}; \mathbf{W}_{ox}]$. $\mathbf{W}_i = [\mathbf{W}_{ih}; \mathbf{W}_{ix}]$. $\mathbf{W}_g = [\mathbf{W}_{gh}; \mathbf{W}_{gx}]$. $\mathbf{W}_f = [\mathbf{W}_{fh}; \mathbf{W}_{fx}]$.

---

**Notation 1**

$$\mathbf{\Delta}_t^c = diag[\mathbf{o}_t \odot (1 - \tanh^2(\mathbf{c}_t))]$$

$$\mathbf{\Delta}_t^o = diag[\mathbf{o}_t \odot (1 - \mathbf{o}_t) \odot \tanh(\mathbf{c}_t)]$$

$$\mathbf{\Delta}_t^f = diag[\mathbf{f}_t \odot (1 - \mathbf{f}_t) \odot \mathbf{c}_{t-1}]$$

$$\mathbf{\Delta}_t^i = diag[\mathbf{i}_t \odot (1 - \mathbf{i}_t) \odot \mathbf{g}_t]$$

$$\mathbf{\Delta}_t^g = diag[(1 - \mathbf{g}_t^2) \odot \mathbf{i}_t]$$

*For any $\star \in \{f, g, o, i\}$, define $\mathbf{E}^\star(w)$ to be a matrix of size $dim(\mathbf{h}_t) \times dim([\mathbf{h}_t; \mathbf{x}_t])$. We set all the elements of this matrix to 0s if if $w$ is not an element of $\mathbf{W}_\star$. Further, if $w = (\mathbf{W}_\star)_{kl}$, then $(\mathbf{E}^\star(w))_{kl} = 1$ and $(\mathbf{E}^\star(w))_{k'l'} = 0$ for all $(k', l') \neq (k, l)$.*

$$\psi_t = \mathbf{\Delta}_t^f \mathbf{W}_{fh} + \mathbf{\Delta}_t^g \mathbf{W}_{gh} + \mathbf{\Delta}_t^i \mathbf{W}_{ih}$$

$$\tilde{\psi}_t = \mathbf{\Delta}_t^o \mathbf{W}_{oh} + \mathbf{\Delta}_t^c \psi_t$$

$$\mathbf{k}_t = \left( \mathbf{\Delta}_t^f \mathbf{E}^f(w) + \mathbf{\Delta}_t^g \mathbf{E}^g(w) + \mathbf{\Delta}_t^i \mathbf{E}^i(w) \right) \cdot [\mathbf{h}_{t-1}, \mathbf{x}_t]^T$$

$$\tilde{\mathbf{k}}_t = \mathbf{\Delta}_t^o \mathbf{E}^o(w) \cdot [\mathbf{h}_{t-1}, \mathbf{x}_t]^T + \mathbf{\Delta}_t^c \mathbf{k}_t$$

$$\mathbf{F}_t = diag(\mathbf{f}_t)$$

$$\tilde{\mathbf{F}}_t = \mathbf{\Delta}_t^c diag(\mathbf{f}_t)$$

---

**Lemma 1** *Let us assume $w$ is an entry of the matrix $\mathbf{W}_f, \mathbf{W}_i, \mathbf{W}_g$ or $\mathbf{W}_o$, then*

$$\frac{d\mathbf{f}_t}{dw} = diag[\mathbf{f}_t \odot (1 - \mathbf{f}_t)] \cdot \left( \mathbf{W}_{fh} \cdot \frac{d\mathbf{h}_{t-1}}{dw} + \mathbf{E}^f(w) \cdot [\mathbf{h}_{t-1}, \mathbf{x}_t]^T \right)$$

$$\frac{d\mathbf{o}_t}{dw} = diag[\mathbf{o}_t \odot (1 - \mathbf{o}_t)] \cdot \left( \mathbf{W}_{oh} \cdot \frac{d\mathbf{h}_{t-1}}{dw} + \mathbf{E}^o(w) \cdot [\mathbf{h}_{t-1}, \mathbf{x}_t]^T \right)$$

$$\frac{d\mathbf{i}_t}{dw} = diag[\mathbf{i}_t \odot (1 - \mathbf{i}_t)] \cdot \left( \mathbf{W}_{ih} \cdot \frac{d\mathbf{h}_{t-1}}{dw} + \mathbf{E}^i(w) \cdot [\mathbf{h}_{t-1}, \mathbf{x}_t]^T \right)$$

$$\frac{d\mathbf{g}_t}{dw} = diag[(1 - \mathbf{g}_t^2)] \cdot \left( \mathbf{W}_{gh} \cdot \frac{d\mathbf{h}_{t-1}}{dw} + \mathbf{E}^g(w) \cdot [\mathbf{h}_{t-1}, \mathbf{x}_t]^T \right)$$

***Proof*** *By chain rule of total differentiation,*

$$\frac{d\mathbf{f}_t}{dw} = \frac{\partial \mathbf{f}_t}{\partial w} + \frac{\partial \mathbf{f}_t}{\partial \mathbf{h}_{t-1}} \frac{d\mathbf{h}_{t-1}}{dw}$$

*We note that,*

$$\frac{\partial \mathbf{f}_t}{\partial w} = diag[\mathbf{f}_t \odot (1 - \mathbf{f}_t)] \cdot \mathbf{E}^f(w) \cdot [\mathbf{h}_{t-1}, \mathbf{x}_t]^T$$

*and,*

$$\frac{\partial \mathbf{f}_t}{\partial \mathbf{h}_{t-1}} = diag[\mathbf{f}_t \odot (1 - \mathbf{f}_t)] \cdot \mathbf{W}_{fh} \cdot \frac{d\mathbf{h}_{t-1}}{dw}$$

*which proves the claim for $\frac{d\mathbf{f}_t}{dw}$. The derivation for $\frac{d\mathbf{o}_t}{dw}, \frac{d\mathbf{i}_t}{dw}, \frac{d\mathbf{g}_t}{dw}$ are similar.*

---

Now let us establish recursive formulas for $\frac{d\mathbf{h}_t}{dw}$ and $\frac{d\mathbf{c}_t}{dw}$, using the above formulas

---

**Corollary 1** *Considering the above notations, we have*

$$\frac{d\mathbf{h}_t}{dw} = \mathbf{\Delta}_t^o \mathbf{W}_{oh} \cdot \frac{d\mathbf{h}_{t-1}}{dw} + \mathbf{\Delta}_t^c \cdot \frac{d\mathbf{c}_t}{dw} + \mathbf{\Delta}_t^o \mathbf{E}^o(w) \cdot [\mathbf{h}_{t-1}, \mathbf{x}_t]^T$$

***Proof*** *Recall that $\mathbf{h}_t = \mathbf{o}_t \odot \tanh(\mathbf{c}_t)$, and thus*

$$\frac{d\mathbf{h}_t}{dw} = \frac{d\mathbf{o}_t}{dw} \odot \tanh(\mathbf{c}_t) + \mathbf{o}_t \odot (1 - \tanh^2(\mathbf{c}_t)) \odot \frac{d\mathbf{c}_t}{dw}$$

*Using the previous Lemma as well as the above notation, we get*

$$\frac{d\mathbf{h}_t}{dw} = diag[\mathbf{o}_t \odot (1 - \mathbf{o}_t)] \cdot \left( \mathbf{W}_{oh} \cdot \frac{d\mathbf{h}_{t-1}}{dw} + \mathbf{E}^o(w) \cdot [\mathbf{h}_{t-1}, \mathbf{x}_t]^T \right) \odot \tanh(\mathbf{c}_t) + \mathbf{o}_t \odot (1 - \tanh^2(\mathbf{c}_t)) \odot \frac{d\mathbf{c}_t}{dw}$$

$$= \mathbf{\Delta}_t^o \mathbf{W}_{oh} \cdot \frac{d\mathbf{h}_{t-1}}{dw} + \mathbf{\Delta}_t^o \mathbf{E}^o(w) \cdot [\mathbf{h}_{t-1}, \mathbf{x}_t]^T + \mathbf{o}_t \odot (1 - \tanh^2(\mathbf{c}_t)) \odot \frac{d\mathbf{c}_t}{dw}$$

$$= \mathbf{\Delta}_t^c \frac{d\mathbf{c}_t}{dw} + \mathbf{\Delta}_t^o \mathbf{W}_{oh} \cdot \frac{d\mathbf{h}_{t-1}}{dw} + \mathbf{\Delta}_t^o \mathbf{E}^o(w) \cdot [\mathbf{h}_{t-1}, \mathbf{x}_t]^T$$

---

**Corollary 2** *Considering the above notations, we have*

$$\frac{d\mathbf{c}_t}{dw} = \mathbf{F}_t \frac{d\mathbf{c}_{t-1}}{dw} + \psi_t \cdot \frac{d\mathbf{h}_{t-1}}{dw} + \mathbf{k}_t$$

***Proof*** *Recall that* $\mathbf{c}_t = \mathbf{f}_t \odot \mathbf{c}_{t-1} + \mathbf{i}_t \odot \mathbf{g}_t$, *and thus*

$$\frac{d\mathbf{c}_t}{dw} = \frac{d\mathbf{f}_t}{dw} \odot \mathbf{c}_{t-1} + \mathbf{f}_t \odot \frac{d\mathbf{c}_{t-1}}{dw} + \frac{d\mathbf{g}_t}{dw} \odot \mathbf{i}_t + \mathbf{g}_t \odot \frac{d\mathbf{i}_t}{dw}$$

*Using the previous Lemma as well as the above notation, we get*

$$\frac{d\mathbf{c}_t}{dw} = diag[\mathbf{f}_t \odot (1 - \mathbf{f}_t)] \cdot \left( \mathbf{W}_{fh} \cdot \frac{d\mathbf{h}_{t-1}}{dw} + \mathbf{E}^f(w) \cdot [\mathbf{h}_{t-1}, \mathbf{x}_t]^T \right) \odot \mathbf{c}_{t-1} + \mathbf{f}_t \odot \frac{d\mathbf{c}_{t-1}}{dw}$$

$$+ diag[(1 - \mathbf{g}_t^2)] \cdot \left( \mathbf{W}_{gh} \cdot \frac{d\mathbf{h}_{t-1}}{dw} + \mathbf{E}^g(w) \cdot [\mathbf{h}_{t-1}, \mathbf{x}_t]^T \right) \odot \mathbf{i}_t$$

$$+ diag[\mathbf{i}_t \odot (1 - \mathbf{i}_t)] \cdot \left( \mathbf{W}_{ih} \cdot \frac{d\mathbf{h}_{t-1}}{dw} + \mathbf{E}^i(w) \cdot [\mathbf{h}_{t-1}, \mathbf{x}_t]^T \right) \odot \mathbf{g}_t$$

$$= \boldsymbol{\Delta}_t^f \mathbf{W}_{fh} \cdot \frac{d\mathbf{h}_{t-1}}{dw} + \boldsymbol{\Delta}_t^f \mathbf{E}^f(w) \cdot [\mathbf{h}_{t-1}, \mathbf{x}_t]^T + \mathbf{f}_t \odot \frac{d\mathbf{c}_{t-1}}{dw}$$

$$+ \boldsymbol{\Delta}_t^g \mathbf{W}_{gh} \cdot \frac{d\mathbf{h}_{t-1}}{dw} + \boldsymbol{\Delta}_t^g \mathbf{E}^g(w) \cdot [\mathbf{h}_{t-1}, \mathbf{x}_t]^T$$

$$+ \boldsymbol{\Delta}_t^i \mathbf{W}_{ih} \cdot \frac{d\mathbf{h}_{t-1}}{dw} + \boldsymbol{\Delta}_t^i \mathbf{E}^i(w) \cdot [\mathbf{h}_{t-1}, \mathbf{x}_t]^T$$

$$= \mathbf{F}_t \frac{d\mathbf{c}_{t-1}}{dw} + \left( \boldsymbol{\Delta}_t^f \mathbf{W}_{fh} + \boldsymbol{\Delta}_t^g \mathbf{W}_{gh} + \boldsymbol{\Delta}_t^i \mathbf{W}_{ih} \right) \cdot \frac{d\mathbf{h}_{t-1}}{dw}$$

$$+ \left( \boldsymbol{\Delta}_t^f \mathbf{E}^g(w) + \boldsymbol{\Delta}_t^g \mathbf{E}^g(w) + \boldsymbol{\Delta}_t^i \mathbf{E}^g(w) \right) \cdot [\mathbf{h}_{t-1}, \mathbf{x}_t]^T$$

$$= \mathbf{F}_t \frac{d\mathbf{c}_{t-1}}{dw} + \psi_t \cdot \frac{d\mathbf{h}_{t-1}}{dw} + \mathbf{k}_t$$

Let us now combine corollary 1 and 2 to get a recursive expression of $\frac{d\mathbf{h}_t}{dw}$ in terms of $\frac{d\mathbf{h}_{t-1}}{dw}$ and $\frac{d\mathbf{c}_{t-1}}{dw}$

**Corollary 3** *Considering the above notations, we have*

$$\frac{d\mathbf{h}_t}{dw} = \tilde{\mathbf{F}}_t \frac{d\mathbf{c}_{t-1}}{dw} + \tilde{\psi}_t \cdot \frac{d\mathbf{h}_{t-1}}{dw} + \tilde{\mathbf{k}}_t$$

***Proof*** *From Corollary 1, we know that*

$$\frac{d\mathbf{h}_t}{dw} = \boldsymbol{\Delta}_t^o \mathbf{W}_{oh} \cdot \frac{d\mathbf{h}_{t-1}}{dw} + \boldsymbol{\Delta}_t^c \cdot \frac{d\mathbf{c}_t}{dw} + \boldsymbol{\Delta}_t^o \mathbf{E}^o(w) \cdot [\mathbf{h}_{t-1}, \mathbf{x}_t]^T$$

*Using Corollary 2, we get*

$$\frac{d\mathbf{h}_t}{dw} = \boldsymbol{\Delta}_t^o \mathbf{W}_{oh} \cdot \frac{d\mathbf{h}_{t-1}}{dw} + \boldsymbol{\Delta}_t^c \cdot \left( \mathbf{F}_t \frac{d\mathbf{c}_{t-1}}{dw} + \psi_t \cdot \frac{d\mathbf{h}_{t-1}}{dw} + \mathbf{k}_t \right) + \boldsymbol{\Delta}_t^o \mathbf{E}^o(w) \cdot [\mathbf{h}_{t-1}, \mathbf{x}_t]^T$$

$$= \boldsymbol{\Delta}_t^c \cdot \mathbf{F}_t \frac{d\mathbf{c}_{t-1}}{dw} + (\boldsymbol{\Delta}_t^o \mathbf{W}_{oh} + \psi_t) \cdot \frac{d\mathbf{h}_{t-1}}{dw} + \left( \mathbf{k}_t + \boldsymbol{\Delta}_t^o \mathbf{E}^o(w) \cdot [\mathbf{h}_{t-1}, \mathbf{x}_t]^T \right)$$

$$= \tilde{\mathbf{F}}_t \frac{d\mathbf{c}_{t-1}}{dw} + \tilde{\psi}_t \cdot \frac{d\mathbf{h}_{t-1}}{dw} + \tilde{\mathbf{k}}_t$$

**Theorem 1** *Fix $w$ to be an element of the matrix $\mathbf{W}_{gh}, \mathbf{W}_{fh}, \mathbf{W}_{ih}$ or $\mathbf{W}_{oh}$. Define,*

$$\mathbf{A}_t = \begin{bmatrix} \mathbf{F}_t & \mathbf{0}_n & diag(\mathbf{k}_t) \\ \tilde{\mathbf{F}}_t & \mathbf{0}_n & diag(\tilde{\mathbf{k}}_t) \\ \mathbf{0}_n & \mathbf{0}_n & \mathbf{Id}_n \end{bmatrix} \quad \mathbf{B}_t = \begin{bmatrix} \mathbf{0}_n & \psi_t & \mathbf{0}_n \\ \mathbf{0}_n & \tilde{\psi}_t & \mathbf{0}_n \\ \mathbf{0}_n & \mathbf{0}_n & \mathbf{0}_n \end{bmatrix} \quad \mathbf{z}_t = \begin{bmatrix} \frac{d\mathbf{c}_t}{dw} \\ \frac{d\mathbf{h}_t}{dw} \\ \mathbf{1}_n \end{bmatrix} \tag{9}$$

*Then,*

$$\mathbf{z}_t = (\mathbf{A}_t + \mathbf{B}_t)\mathbf{z}_{t-1}$$

*In other words,*

$$\mathbf{z}_t = (\mathbf{A}_t + \mathbf{B}_t)(\mathbf{A}_{t-1} + \mathbf{B}_{t-1})\dots(\mathbf{A}_2 + \mathbf{B}_2)\mathbf{z}_1$$

*where all the symbols used to define $\mathbf{A}_t$ and $\mathbf{B}_t$ are defined in notation 1.*

**Proof** *By Corollary 2, we get*

$$\frac{d\mathbf{c}_t}{dw} = \mathbf{F}_t \frac{d\mathbf{c}_{t-1}}{dw} + \psi_t \cdot \frac{d\mathbf{h}_{t-1}}{dw} + \mathbf{k}_t$$

$$= \mathbf{F}_t \frac{d\mathbf{c}_{t-1}}{dw} + \psi_t \cdot \frac{d\mathbf{h}_{t-1}}{dw} + diag(\mathbf{k}_t)\mathbf{1}_n$$

$$= \begin{bmatrix} \mathbf{F}_t & \psi_t & diag(\mathbf{k}_t) \end{bmatrix} \cdot \mathbf{z}_{t-1}$$

*Similarly by Corollary 3, we get*

$$\frac{d\mathbf{h}_t}{dw} = \tilde{\mathbf{F}}_t \frac{d\mathbf{c}_{t-1}}{dw} + \tilde{\psi}_t \cdot \frac{d\mathbf{h}_{t-1}}{dw} + \tilde{\mathbf{k}}_t$$

$$= \tilde{\mathbf{F}}_t \frac{d\mathbf{c}_{t-1}}{dw} + \tilde{\psi}_t \cdot \frac{d\mathbf{h}_{t-1}}{dw} + diag(\tilde{\mathbf{k}}_\mathbf{t})\mathbf{1}_n$$

$$= \begin{bmatrix} \tilde{\mathbf{F}}_t & \tilde{\psi}_t & diag(\tilde{\mathbf{k}}_t) \end{bmatrix} \cdot \mathbf{z}_{t-1}$$

*Thus we have*

$$\mathbf{z}_t = \begin{bmatrix} \mathbf{F}_t & \psi_t & diag(\mathbf{k}_t) \\ \tilde{\mathbf{F}}_t & \tilde{\psi}_t & diag(\tilde{\mathbf{k}}_\mathbf{t}) \\ \mathbf{0}_n & \mathbf{0}_n & \mathbf{Id}_n \end{bmatrix} \cdot \mathbf{z}_{t-1} = (\mathbf{A}_t + \mathbf{B}_t) \cdot \mathbf{z}_{t-1} \tag{10}$$

*Applying this formula recursively proves the claim.*

Note: Since $\mathbf{A}_t$ has $\mathbf{0}_n$'s in the second column of the block matrix representation, it ignores the contribution of $\mathbf{z}_t$ coming from $\mathbf{h}_{t-1}$, whereas $\mathbf{B}_t$ (having non-zero block matrices only in the second column of the block matrix representation) only takes into account the contribution coming from $\mathbf{h}_{t-1}$. Hence $\mathbf{A}_t$ captures the contribution of the gradient coming from the cell state $\mathbf{c}_{t-1}$.

## C   Derivation of Back-propagation Equation for LSTM with $h$-detach

**Theorem 2**  *Let* $\mathbf{z}_t = [\frac{d\mathbf{c}_t}{dw}^T; \frac{d\mathbf{h}_t}{dw}^T; \mathbf{1}_n^T]^T$ *and* $\tilde{\mathbf{z}}_t$ *be the analogue of* $\mathbf{z}_t$ *when applying* $h$-detach *with probability* $p$ *during back-propagation. Then,*

$$\tilde{\mathbf{z}}_t = (\mathbf{A}_t + \xi_t \mathbf{B}_t)(\mathbf{A}_{t-1} + \xi_{t-1}\mathbf{B}_{t-1})\ldots(\mathbf{A}_2 + \xi_2 \mathbf{B}_2)\tilde{\mathbf{z}}_1$$

*where* $\xi_t, \xi_{t-1}, \ldots, \xi_2$ *are i.i.d. Bernoulli random variables with probability* $p$ *of being 1,* $\mathbf{A}_t$ *and* $\mathbf{B}_t$ *and are same as defined in theorem 1.*

**Proof** *Replacing* $\frac{\partial}{\partial \mathbf{h}_{t-1}}$ *by* $\xi_t \frac{\partial}{\partial \mathbf{h}_{t-1}}$ *in lemma 1 and therefore in Corollaries 2 and 3, we get the following analogous equations*

$$\frac{d\mathbf{c}_t}{dw} = \mathbf{F}_t \frac{d\mathbf{c}_{t-1}}{dw} + \xi_t \psi_t \cdot \frac{d\mathbf{h}_{t-1}}{dw} + \mathbf{k}_t$$

*and*

$$\frac{d\mathbf{h}_t}{dw} = \tilde{\mathbf{F}}_t \frac{d\mathbf{c}_{t-1}}{dw} + \xi_t \tilde{\psi}_t \cdot \frac{d\mathbf{h}_{t-1}}{dw} + \tilde{\mathbf{k}}_t$$

*Similarly as in the proof of previous theorem, we can rewrite*

$$\frac{d\mathbf{c}_t}{dw} = \begin{bmatrix} \mathbf{F}_t & \xi_t \psi_t & diag(\mathbf{k}_t) \end{bmatrix} \cdot \tilde{\mathbf{z}}_{t-1}$$

*and*

$$\frac{d\mathbf{h}_t}{dw} = \begin{bmatrix} \tilde{\mathbf{F}}_t & \xi_t \tilde{\psi}_t & diag(\tilde{\mathbf{k}}_t) \end{bmatrix} \cdot \tilde{\mathbf{z}}_{t-1}$$

*Thus*

$$\tilde{\mathbf{z}}_t = \begin{bmatrix} \mathbf{F}_t & \xi_t \psi_t & diag(\mathbf{k}_t) \\ \tilde{\mathbf{F}}_t & \xi_t \tilde{\psi}_t & diag(\tilde{\mathbf{k}}_t) \\ \mathbf{0}_n & \mathbf{0}_n & \mathbf{Id}_n \end{bmatrix} \cdot \tilde{\mathbf{z}}_{t-1} = \left( \begin{bmatrix} \mathbf{F}_t & \mathbf{0}_n & diag(\mathbf{k}_t) \\ \tilde{\mathbf{F}}_t & \mathbf{0}_n & diag(\tilde{\mathbf{k}}_t) \\ \mathbf{0}_n & \mathbf{0}_n & \mathbf{Id}_n \end{bmatrix} + \xi_t \begin{bmatrix} \mathbf{0}_n & \psi_t & \mathbf{0}_n \\ \mathbf{0}_n & \tilde{\psi}_t & \mathbf{0}_n \\ \mathbf{0}_n & \mathbf{0}_n & \mathbf{0}_n \end{bmatrix} \right) \cdot \tilde{\mathbf{z}}_{t-1}$$

$$= (\mathbf{A}_t + \xi_t \mathbf{B}_t) \cdot \tilde{\mathbf{z}}_{t-1}$$

*Iterating this formula gives,*

$$\tilde{\mathbf{z}}_t = (\mathbf{A}_t + \xi_t \mathbf{B}_t)(\mathbf{A}_{t-1} + \xi_{t-1}\mathbf{B}_{t-1})\ldots(\mathbf{A}_3 + \xi_3 \mathbf{B}_3)\tilde{\mathbf{z}}_2$$

**Corollary 4**
$$\mathbb{E}[\tilde{\mathbf{z}}_t] = (\mathbf{A}_t + p\mathbf{B}_t)(\mathbf{A}_{t-1} + p\mathbf{B}_{t-1})\ldots(\mathbf{A}_3 + p\mathbf{B}_3)\tilde{\mathbf{z}}_2$$

*It suffices to take the expectation both sides, and use independence of* $\xi_t$*'s.*

