# OpenReview forum: "h-detach: Modifying the LSTM Gradient Towards Better Optimization"
_ICLR.cc/2019/Conference_

### Official Review · AnonReviewer1 · 2018-10-31
**Intriguing results. But don't similar methods achieve similar things with similar mechanisms?**

**Rating:** 7
**Confidence:** 5

**Review:**

The results are intriguing. However, similar methods like BN-LSTM [3] and Variational RNNs [4] achieve arguably the same with very similar mechanisms. We do not think they can be considered as orthogonal. This should be addressed by the authors. Also, hard long-term experiments like sequentially predicting pixels (like through MDLSTM-based PixelRNN) or language modelling should be favoured over short sentence image captions.

It is possible that we will improve our ratings once our concerns are addressed.

Paper Summary:

The authors claim that the gradient along the computational path that goes through the cell state (the linear temporal path or A gradient) of an LSTM carries information about long-term dependencies. Those gradients can be corrupted by the gradient of all other computational paths (i.e. the B gradient). They claim that this makes it hard to learn long-term dependencies and has, therefore, significant negative effects on the convergence speed, training stability, and generalisation performance. They propose a method called h-detach and run experiments on the delayed copy task, sequential MNIST, permuted sequential MNIST (pMNIST), and caption generation on the MS COCO dataset. All show either somewhat improved performance or much more stable learning curves. At every step, h-detach randomly drops all gradients that flow through the h of the standard LSTM, the B gradients, and only keeps the ones from the linear temporal path, the A gradients. Experiments also suggest that the A gradients carry more long-term information than B gradients and that LSTMs with h-detach do not need gradient clipping for successful training.

Positive:

The paper is written clearly. It is well structured and well motivated. H-detach is simple, effective, and somewhat novel (see below). Experiments indicate that its main benefit is training stability as well as minor performance improvements.

Negative:

We are not sure how significant these results are for the following reasons:

- MS COCO image caption generation is the only more challenging dataset, but it seems a bit misplaced as it has very short sentences, while the authors motivate their work through a focus on long-term dependencies. Why not apply h-detach to a language model such as [1] with official online implementations, e.g., [2]. A setting with PixelRNN [6] based on MD-LSTM [7] would also be a great testbed for h-detach.

- The purpose of h-detach is to scale down the B gradients. However, methods which apply e.g. BatchNorm to the hidden state learn a scale parameter which could be learned by the network explicitly. For the backward pass, this has the effect of scaling down the B gradient. Consider e.g. [3] which also achieves similar training stability on sequential MNIST and pMNIST with little overhead.

- Another very related method is [4] which properly applies a random dropout mask over the recurrent inputs that is shared across timesteps of an RNN. We think that h-detach is essentially achieving the same in a similar way.

Problems with Introduction and Related Work Section:

- The vanishing gradient problem was first described by Hochreiter in 1991 [5] (not by Bengio in 1994).

- Intro mentions GRU as if it was separate from LSTM. Clarify that GRU is essentially a variant of vanilla LSTM with forget gates [8]. Since one gate is missing, GRU is less powerful than the original LSTM [9].

[1] Zaremba et al. "Recurrent neural network regularization." arXiv:1409.2329 (2014).
[2] https://www.tensorflow.org/tutorials/sequences/recurrent
[3] Cooijmans et al. "Recurrent batch normalization." arXiv:1603.09025 (2016).
[4] Gal et al. "A theoretically grounded application of dropout in recurrent neural networks." NIPS 2016.
[5] Hochreiter, Sepp. "Untersuchungen zu dynamischen neuronalen Netzen." Diploma thesis, TUM (1991)
[6] Oord et al. "Pixel recurrent neural networks." arXiv preprint arXiv:1601.06759 (2016).
[7] Graves et al. "Multi-Dimensional Recurrent Neural Networks" arXiv preprint arXiv:0705.2011 (2011).
[8] Gers et al. “Learning to Forget: Continual Prediction with LSTM.“ Neural Computation, 12(10):2451-2471, 2000.
[9] Weiss et al. On the Practical Computational Power of Finite Precision RNNs for Language Recognition. arXiv:1805.04908.


Comments after rebuttal:

The  paper has clearly improved.

It leaves a few questions open though. For example, it is surprising that h-detach doesn't work on language modelling since Dropout-LSTM and BN-LSTM clearly improve over vanilla LSTM in this case (if not every case). In the new version, the authors only reference it in one or two sentences but don't discuss this in detail.

When dropout is mentioned, one should also mention that dropout is a variant of the old stochastic delta rule:

Hanson, S. J. (1990). A Stochastic Version of the Delta Rule, PHYSICA D,42, 265-272.  See also arXiv:1808.03578

Nevertheless, we now think that this is a very interesting LSTM regularization paper that people who study this field should probably know. We are increasing the score by 2 points!

---

> ### Author Response · Authors · 2018-11-11
> **Rebuttal**
>
> We highly appreciate your constructive comments and the missing citations you provided.
>
> Thank you for bringing up the point that image captioning does not fit the profile of a task involving long term dependencies. We believe the reason why our method leads to improvements for the image captioning task is that the gradient components from the cell state path are important for this task. As our theoretical analysis of h-detach shows, it prevents these components from getting suppressed compared with the gradient components from the h-state paths. Since the obvious target for our method were tasks involving long term dependencies, we use it as our main pitch. We have revised the paper with these comments.
> Also, we did try our method on language modeling tasks but we did not find any benefit in these cases. We have added this detail in the discussion section 5 of the revised version.
>
> Recurrent batch normalization is indeed beneficial for training LSTMs. However, as the reviewer pointed out, it adds computation overhead and its implementation is quite involved (and also adds dependence on mini-batch statistics). Our method on the other hand reduces computation needed for vanilla LSTM and is very simple to implement while improving the convergence speed and robustness over traditional LSTM updates.
>
> For a discussion on difference between dropout and h-detach, please see our reply to AnonReviewer 2. We understand that the version of dropout referred by the reviewer is different from the original dropout technique. But the difference we have stated applies to this version of dropout as well.
>
> We thank the reviewer for point out the earlier manuscript that noticed the vanishing gradient problem. We, the main authors of the paper, were not aware of this, especially given the manuscript is not in English. We have cited the thesis at all places in the paper when referring to vanishing gradients in the revised version (introduction and related work sections).
>
> We have changed the sentence saying GRU is a variant of LSTM with forget gates. We have also pointed out that LSTMs are more powerful compared with GRUs along with the citation mentioned by the reviewer. These changes have been added in the introduction section of the revised version.
>
> Finally, we would like to point out that the main benefits of our simple algorithm for modifying the LSTM update direction (for which we provide theoretical analysis) are that it leads to improvements in convergence speed, robustness to seed and learning rate, and generalization as shown in Fig. 2,3 and 6.
>
> We hope we have addressed your concerns.

---

### Official Review · AnonReviewer3 · 2018-11-02

**Rating:** 6
**Confidence:** 3

**Review:**

In this paper, the authors propose a simple modification to the training process of the LSTM. The goal is to facilitate gradient propagation along cell states, or the "linear temporal path". It blocks the gradient propagation of hidden states with a probability of $1-p$ independently. The proposed method is evaluated on the copying task, sequential MNIST task, and image captioning tasks. The performance is sightly boosted on those tasks.

The paper is well-written. The h-detach method is very simple and easy to implement. It seems novel in dealing with the trainability issue with recurrent networks. Since LSTM is very commonly used, if the method is proved to be effective on other tasks, it will potentially benefit a large portion of the community. However, the reviewer thinks the paper is not sufficiently motivated and the quality of the paper could be further improved by conducting a more thorough analysis of the proposed method, and discussing the connection with other existing methods.

As the motivation of the work, the authors seem to claim that if the magnitude of $B_t$ is much bigger that $A_t$, then the backpropagation will be problematic. Is there any theoretical or empirical support of this claim?

In order to damp the gradient component of $B_t$, it does not have to be stochastic. Can we simply multiply the matrix $B_t$ by a constant factor $p$ during backpropagation? Or regularize the weights $W_{*h}$ to be small so that $\phi_t$ and $\tilde\phi_t$ are small?

It would be interesting to study the effect of the probability $p$ and to suggest an "optimal" choice of $p$, either theoretically or empirically. Is it still possible to train the model with a very small $p$?

The h-detach method seems to have a flavor of dropout, but the "dropout" only happens during backpropagation. The design goal also resembles the peephole LSTM, that is to disentangle the cell state and the hidden state. Is there any possible connections between the proposed method and the dropout and peephole LSTM?

The reviewer understands that a one percent difference in the accuracy on MNIST is probably not very meaningful, but it seems that the SOTA performance on pMNIST is at least 94.1% [1].

[1] Scott Wisdom, Thomas Powers, John Hershey, Jonathan Le Roux, and Les Atlas. Full-capacity unitary recurrent neural networks. NIPS, 2016.

---

> ### Author Response · Authors · 2018-11-11
> **Rebuttal**
>
> We thank you for your insightful comments.
>
> We provided a theoretical analysis showing that the gradient through the cell state (A_t) gets suppressed when the gradient through the h-states (B_t) are larger in magnitude (theorem 1 and 2 and the discussions around them).
> We indeed have provided empirical support for this claim. In ablation studies, we show that blocking the gradients through the cell states result in extremely poor performance of LSTMs on both pixel MNIST and copying task. On the other hand, the use of our method on these tasks which stochastically blocks the gradients through the h-states of the LSTM results in faster convergence. In the former case, the theory guarantees that B_t overwhelms A_t, while in the latter case A_t becomes comparable to B_t.
>
> Your insights are perfectly correct. In order to damp the gradient components of B_t, we can indeed multiply B_t by a constant factor during back propagation or regularize the weights of the h-state path to be small. We have added these remarks as future work in the revised paper in section 5.
>
> For MNIST task, when training a model with a very small p=0.001, the convergence was quite slow and the final model was worse than baseline. Further, in our internal experiments, we tried detach probability p with values 0.1, 0.25, 0.4, 0.5, 0.75 and 0.9. We found that the values between 0.25 and 0.5 usually had best performance and so we used values in this interval for our hyper-parameter search.
>
> Peephole LSTM makes all the gates depend on previous cell state in addition to h-state and the current input. The computational graph of peephole LSTM will have an edge pointing from C(t-1) to h(t) in Fig. 1 of our paper. Hence at least intuitively, we believe it will not be able to prevent the gradient component through the cell state path from being suppressed because the gradient component through the other paths will still grow polynomially as the magnitude of recurrent weights grow.
>
> Regarding the improvement in SOTA, we believe that the main benefit of our method is improvement in training stability, convergence and robustness to seed for tasks where the cell states carry important information about the task. For instance, it has been shown that recurrent networks are sensitive to the randomness in initialization ("seed" in coding terminology) [1]. In our paper, we reported experiments on various seeds and learning rate showing these aforementioned benefits (Fig. 2,3,4,6,8 in the revised version). Additionally, our goal was not to compete with existing SOTA algorithms which we believe may also benefit from our method when used in conjunction. Our goal was to rather to investigate and alleviate the source of the problem that makes the training of LSTMs unstable and sensitive to seed when training on tasks where the cell states carry important information (such as tasks involving long term dependencies).
>
> [1] On the State of the Art of Evaluation in Neural Language Models

---

### Official Review · AnonReviewer2 · 2018-11-07
**Interesting but there are some technical details missing**

**Rating:** 5
**Confidence:** 4

**Review:**

The author introduces a simple stochastic algorithm (h-detach) that is specific to LSTM optimization and targeted towards addressing this problem. Specifically, the authors show that when the LSTM weights are large, the gradient components through the linear path (cell state) in the LSTM computational graph get suppressed. Based on the hypothesis that these components carry information about long term dependencies (which we show empirically), their suppression can prevent the LSTM from capturing them. Our algorithm prevents the gradients flowing through this path from getting suppressed, thus allowing the LSTM to capture such dependencies better. The experimental results show that the proposed algorithm appears to be effective. Some detailed comments are listed as follows,

1 The h-detach algorithm seems to be the dropout technology. However, the authors did not discuss the relation or difference between the proposed h-detach algorithm and the dropout technology.

2 The proposed method can transfer the positive knowledge. However, for the transfer learning, one concerned and important issue is that some negative knowledge information can be also transferred. So how to avoid the negative transferring? Some necessary discussions about this should be given in the manuscript.

2 There are many grammar errors in the current manuscript. The authors are suggested to improve the English writing.

---

> ### Author Response · Authors · 2018-11-11
> **Rebuttal**
>
> Thank you for your comments.
>
> Given the superficial similarity, we agree that it warrants a discussion between dropout and our proposed method. The two methods are fundamentally different. Dropout randomly masks the hidden units of a network during the forward pass. Therefore, a common view of dropout is training an ensemble of networks. On the other hand, our method does not mask the hidden units during the forward pass. It instead randomly blocks the gradient component through the h-states (and not cell state, so we block a specific component instead of randomly choosing any component) of the LSTM only during the backward pass and our method does not change the output of the network during forward pass. Our theoretical analysis shows the precise behavior of our method: the effect of this operation is that it changes the update direction used for descent which prevents the gradient components through the cell state path from being suppressed (which we show are important for tasks involving longer term dependencies). We have added this discussion in the revised version in section 5.
>
> Transfer copy task is a commonly used benchmark task for evaluating how good a recurrent model is at retaining information over large time scales. Therefore we report numbers on this task purely for this reason. Our goal and the proposed method has nothing to do with transfer learning otherwise.
>
> We would also like to point out that the main benefits of our algorithm which modifies the LSTM update direction (for which we provide theoretical analysis) are that it leads to improvements in convergence speed, robustness to seed and learning rate, and generalization compared to the usual LSTM updates.
>
> We have revised our manuscript. We hope to have addressed your concerns.

---

### Public Comment · (anonymous) · 2019-01-01
**Interesting approach and nice experimentation. Can have more comprehensive related work.**

The simplicity of the method and its effectiveness are very impressive and evaluation on multiple datasets seems to validate the claims.

The only concern I have is about the related work on the stability of RNN training. After the Arjovsky et al., (ICML 2016), there have been many papers on its lines and have improved the techniques even further and the latest being Zhang et al (ICML 2018). The other works on Unitary RNNs include Wisdom et al., (NeurIPS 2016), Mhammedi et al., (ICML 2017), Jing et al., (ICML 2017), Vorontsov et al., (ICML 2017), Jose et al., (ICML 2018). Another slightly orthogonal work (moving away from unitary methods on the stability of RNNs seems to be Kusupati et al., (NeurIPS 2018).

Adding the relevant literature (from above) about stabilizing gradients for RNN training for better and faster optimization will make the related work much more comprehensive, given they fall into the similar class as the current paper.

Thanks.

---

> ### Author Response · Authors · 2019-01-02
> **Thank you!**
>
> Thank you for the references. We will review and add them in our final version.

---

> > ### Public Comment · ~Shuai_Li5 · 2019-02-01
> > **related work on solving gradient exploding and vanishing problems**
> >
> > In addition to the uRNN series of works, the recent IndRNN (Independently recurrent neural network) also addresses the gradient exploding and vanishing problems. Experiments have also shown its effective performance in solving problems concerning long-term dependency (even up to 5000 timesteps). Also it shows a great advantage in constructing deep RNN networks (easily over 20 layers).
> > [1] S. Li, W. Li, C. Cook, C. Zhu, Y. Gao, “Independently Recurrent Neural Network (IndRNN): Building A Longer and Deeper RNN,” IEEE Conference on Computer Vision and Pattern Recognition, Salt Lake City, Jun. 18-22, 2018.
> >
> >
> > Thanks.

---

> > > ### Author Response · Authors · 2019-02-01
> > > **Citation is already present**
> > >
> > > Citation to this paper was added in our final submission.

---

### Public Comment · ~Aniket_Rajiv_Didolkar1 · 2019-01-07
**Reproducibility study of h-detach:Modifying the LSTM Gradient Towards Better Optimization**

As a part of the ICLR 2019 reproducibility challenge, we worked to reproduce the results of this paper (h-detach). The code for reproducing the results was provided by the authors. We perform some of the experiments mentioned in the paper. A link to the full report and the codebase can be found at the end of this message.

The authors tackle the exploding and vanishing gradient problem(EVGP)  in the LSTM. The authors say that when the weights of the LSTM are large, due to their repeated multiplication, the gradients through the cell state path get suppressed. The authors empirically prove that this path carries information about long-term dependencies. The authors have also provided theoretical proof of their claim.

The authors have proposed a stochastic algorithm to mitigate the above mentioned problems. The main motive of the authors is to prove that training an LSTM with h-detach results in faster convergence and more stability during training. We performed experiments for the copying task, sequential mnist task and our results were able to confirm the claims made by the author. We also performed the experiments mentioned in the ablation study and were able to get similar results as the authors.

The authors have given information on how they chose the detach probability for the sequential mnist task. They have also mentioned, in a reply to AnonReviewer3 below, that they tried different values to of the probability and found the values between 0.25 and 0.50 work best. They have sufficiently experimented with their algorithm on different learning rates and seeds. It would also be interesting to see how the performance of h-detach is affected by a change in batch size.

In conclusion, we have validated the claims of the authors through our experiments that using h-detach stabilizes training, leads to faster convergence and is robust to different seeds and learning rates.

codebase: https://github.com/dido1998/h-detach
report: https://drive.google.com/drive/folders/1EtEYBS0LRRGsCwhxiXwfDY2ndsgCJPpc?usp=sharing

---

### Meta-Review · Area_Chair1 · 2018-12-09
**a simple but well motivated trick for stabilizing LSTM optimization**

**Confidence:** 5
**Recommendation:** Accept (Poster)

**Metareview:**

This paper presents a method for preventing exploding and vanishing gradients in LSTMs by stochastically blocking some paths of the information flow (but not others). Experiments show improved training speed and robustness to hyperparameter settings.

I'm concerned about the quality of R2, since (as the authors point out) some of the text is copied verbatim from the paper. The other two reviewers are generally positive about the paper, with scores of 6 and 7, and R1 in particular points out that this work has already had noticeable impact in the field. While the reviewers pointed out some minor concerns with the experiments, there don't seem to be any major flaws. I think the paper is above the bar for acceptance.